# Acute intermittent hypoxia enhances corticospinal synaptic plasticity in humans

Lasse Christiansen[1], MA Urbin[1], Gordon S Mitchell[2,3,4], Monica A Perez[1]*

[1]Department of Neurological Surgery, The Miami Project to Cure Paralysis, University of Miami, Miami, United States; [2]Center for Respiratory Research and Rehabilitation, University of Florida, Gainesville, United States; [3]Department of Physical Therapy, University of Florida, Gainesville, United States; [4]McKnight Brain Institute, University of Florida, Gainesville, United States

**Abstract** Acute intermittent hypoxia (AIH) enhances voluntary motor output in humans with central nervous system damage. The neural mechanisms contributing to these beneficial effects are unknown. We examined corticospinal function by evaluating motor evoked potentials (MEPs) elicited by cortical and subcortical stimulation of corticospinal axons and the activity in intracortical circuits in a finger muscle before and after 30 min of AIH or sham AIH. We found that the amplitude of cortically and subcortically elicited MEPs increased for 75 min after AIH but not sham AIH while intracortical activity remained unchanged. To examine further these subcortical effects, we assessed spike-timing dependent plasticity (STDP) targeting spinal synapses and the excitability of spinal motoneurons. Notably, AIH increased STDP outcomes while spinal motoneuron excitability remained unchanged. Our results provide the first evidence that AIH changes corticospinal function in humans, likely by altering corticospinal-motoneuronal synaptic transmission. AIH may represent a novel noninvasive approach for inducing spinal plasticity in humans.
DOI: https://doi.org/10.7554/eLife.34304.001

*For correspondence:
perezmo@miami.edu (MAP)

Competing interest: The authors declare that no competing interests exist.

## Introduction

Brief exposures to hypoxic air interspersed with periods of breathing ambient room air, known as acute intermittent hypoxia (AIH), impacts the respiratory, cardiovascular, immune, metabolic, bone and nervous systems (*Dale et al., 2014*; *Gonzalez-Rothi et al., 2015*). During recent years, an increasing number of studies support the view that AIH also affects the damaged central nervous system, triggering recovery of motor function in humans with partial paralysis due to spinal cord injury (*Trumbower et al., 2012*; *Lynch et al., 2017*; *Navarrete-Opazo et al., 2017a, b*). AIH represents a promising and safe strategy to supplement conventional neurorehabilitation (*Navarrete-Opazo and Mitchell, 2014*). However, the neural mechanisms contributing to the beneficial effects of AIH in the human motor system remain unknown.

Studies in animal models have demonstrated that single and multiple exposures to AIH cause phrenic motor facilitation (*Baker and Mitchell, 2000*; *Golder and Mitchell, 2005*) and increase expression of brain-derived neurotrophic factor (BDNF) in respiratory motor neurons (*Baker-Herman et al., 2004*; *Satriotomo et al., 2012*). Changes in the respiratory system are reflected as serotonin-dependent synaptic plasticity in the spinal cord (*Bach and Mitchell, 1996*; *Fuller et al., 2001*; *Baker-Herman and Mitchell, 2002*). Other studies showed that AIH elicits plasticity in neural systems not directly linked to the respiratory system. For example, exposure to AIH transiently increases sympathetic nerve activity (*Dick et al., 2007*; *Xing and Pilowsky, 2010*) and connectivity of spinal interneurons projecting to midthoracic motoneurons (*Streeter et al., 2017*). A critical question is if AIH can modulate activity in descending motor pathways in intact humans and its mechanisms of action.

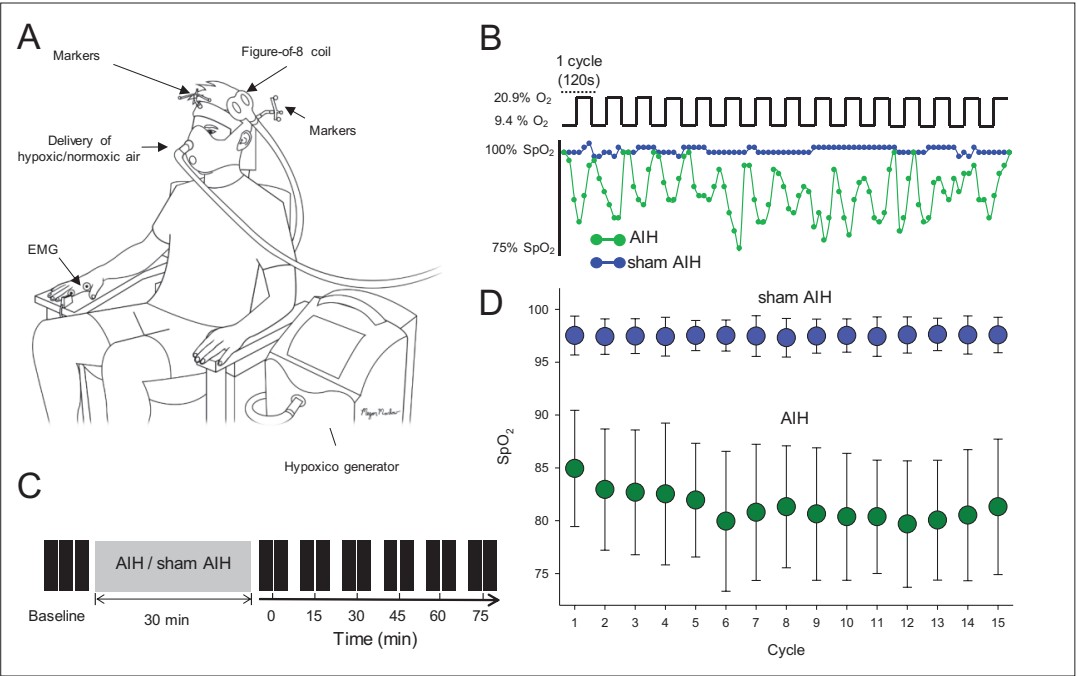

**Figure 1.** Experimental setup. (**A**) Participants were seated in a customized chair during acute intermittent hypoxia (AIH) and sham AIH protocols using the Hypoxico Inc. (EVEREST SUMMIT II, New York). During electrophysiological outcomes using transcranial magnetic stimulation (TMS) the coil was monitored by markers using a frameless stereotaxic system and electromyographic (EMG) activity was recorded from the first dorsal interosseous (FDI) muscle. (**B**) SpO2 from a representative participant during the AIH (green) and sham AIH (blue) protocols. The AIH protocol consisted of 15 cycles of 1 min of inspiring ambient air (20.9% O2) with 1 min of hypoxic air (9.4% O2). (**C**) Timeline of the experimental protocol. Electrophysiological outcomes were measured before (baseline), immediately after (0) and 15, 30, 45, 60 and 75 min each after protocol. (**D**) Group data (n = 19) showing the SpO2 observed during the 15 cycles for the AIH (green circles) and sham AIH (blue circles) protocols. Error bars denote SDs.

DOI: https://doi.org/10.7554/eLife.34304.006

The corticospinal tract is a major descending motor pathway contributing to the control of voluntary movement (*Lemon, 2008*). Plasticity in the corticospinal tract contributes to the recovery of motor function in individuals with a variety of motor disorders (*Ridding and Rothwell, 2007*; *Oudega and Perez, 2012*). Serotonin (*Batsikadze et al., 2013*; *Kuo et al., 2016*) and BDNF (*Fritsch et al., 2010*), which are both affected by AIH, contribute to plasticity in the corticospinal pathway. Further, repetitive AIH enhances BDNF immunoreactivity in primary motor cortex of rats, especially around pyramidal cells of layer V (*Satriotomo et al., 2016*). We hypothesized that a single session of AIH enhances corticospinal function in intact humans. Since corticospinal transmission depends on the strength of synaptic connections between corticospinal drive and spinal motoneurons (*Taylor and Martin, 2009*; *Bunday and Perez, 2012*) and AIH induces spinal cord plasticity (*Dale et al., 2014*) we predicted that changes in spinal synapses contribute to AIH-mediated effects in corticospinal transmission.

To test our hypotheses we examined motor responses elicited by cortical and subcortical stimulation of corticospinal axons and intracortical inhibition and facilitation in an intrinsic finger muscle before and after 30 min of AIH or sham AIH (*Figure 1*). We also tested spinal plasticity and motoneuron excitability.

## Results

### Motor evoked potentials (MEPs) elicited by transcranial magnetic stimulation (TMS)

*Figure 2* illustrates traces of MEPs elicited by TMS over the primary motor cortex in the FDI muscle from a representative subject during both protocols. Note that the MEP amplitude increased after

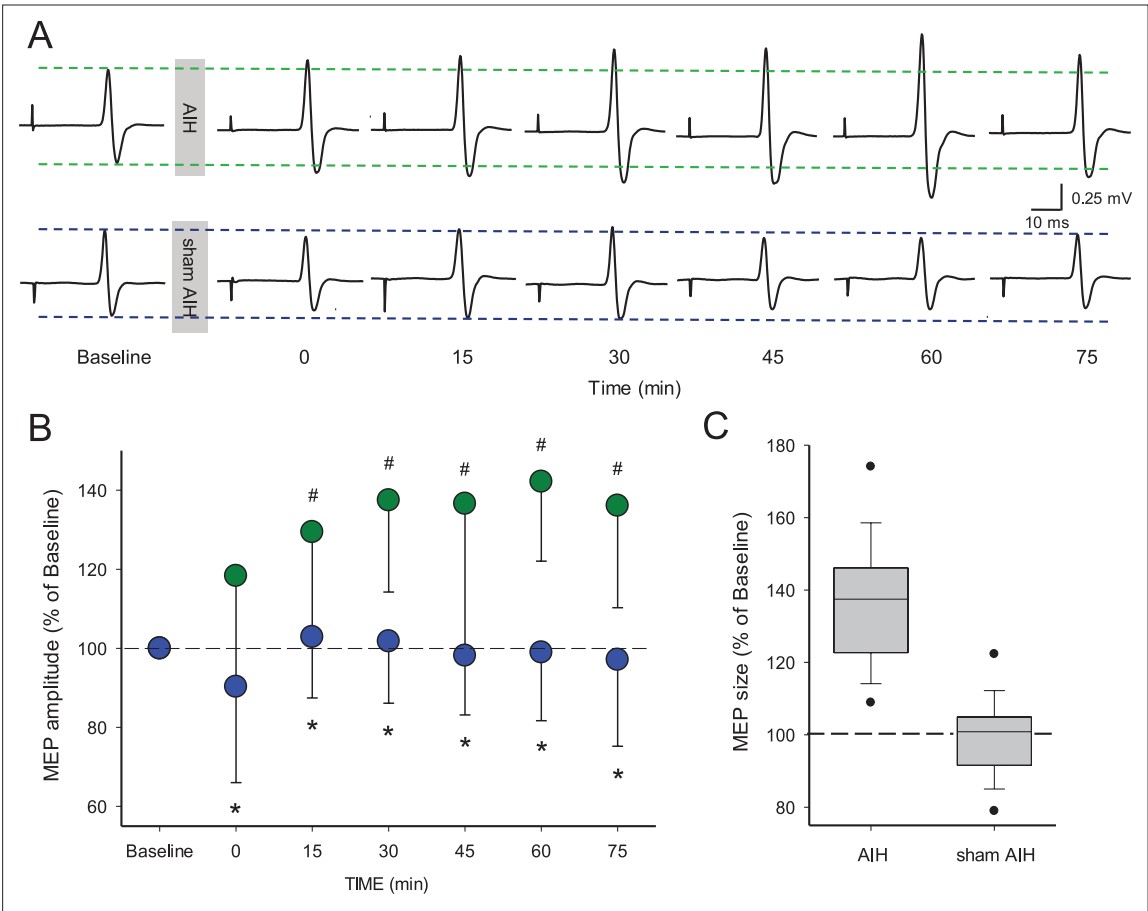

**Figure 2.** Motor evoked potentials (MEPs) elicited by TMS. (**A**) Raw MEP traces elicited by TMS in the FDI muscle in a representative participant before and after the AIH (upper traces) and sham AIH (lower traces) protocols. Each waveform represents the average of 20 MEPs before and after each protocol. (**B**) Group data (n = 19) showing normalized MEP amplitudes before and after the AIH (green circles) and sham AIH (blue circles) protocols. The abscissa shows the time of measurements (baseline, immediately after (0), 15, 30, 45, 60 and 75 min after each protocol). The ordinate shows the MEP amplitude as a percentage of the baseline MEP (% of Baseline) for each protocol. (**C**) Graph shows box-plots group normalized MEP data at baseline (dotted line) and after the AIH and sham AIH protocols from 15-75 minutes averaged (n = 19). The abscissa shows the baseline and protocols tested (AIH and sham AIH) and the ordinate shows the MEP amplitude as a percentage of the baseline MEP (% of Baseline) for each protocol. Whiskers illustrate 5th and 95th percentiles and dots represent minimal and maximal values. Error bars indicate SDs. *p<0.05, comparison between protocols. #p<0.001, comparison with the baseline.

DOI: https://doi.org/10.7554/eLife.34304.002

the AIH but not the sham AIH protocol for up to 75 min. Repeated-measures ANOVA revealed an effect of TIME ($F_{3.86,18}$=5.6, p<0.001), GROUP ($F_{1,18}$=80.8, p<0.001) and in their interaction ($F_{3.86,108}$=4.7, p<0.001) on MEP amplitude. Tukey *Post hoc* tests showed that MEP amplitude increased after 15 (29.4 ± 26.2%; p<0.001), 30 (37.4 ± 23.2%; p<0.001), 45 (36.5 ± 38.3%; p<0.001), 60 (42.1 ± 20.0%; p<0.001) and 75 (36.0 ± 25.8%; p<0.001) min after AIH compared with baseline (*Figure 2B*). No changes were observed in MEP amplitude after the sham AIH protocol compared with baseline at any of the time point tested (all p=0.10). Note that MEPs were facilitated in all of subjects after the AIH protocol (19/19) when comparing baseline to the average of measurements between 15 and 75 min after AIH. We followed up the effects of AIH in one individual and found out that the MEP amplitude returned to baseline ~120 min after AIH.

## Short-interval intracortical inhibition (SICI) and intracortical facilitation (ICF)

*Figure 3A–B* illustrate raw data from SICI and ICF measurements in a representative subject. Note that the magnitude of SICI and ICF remained unchanged after the AIH protocol compared with baseline.

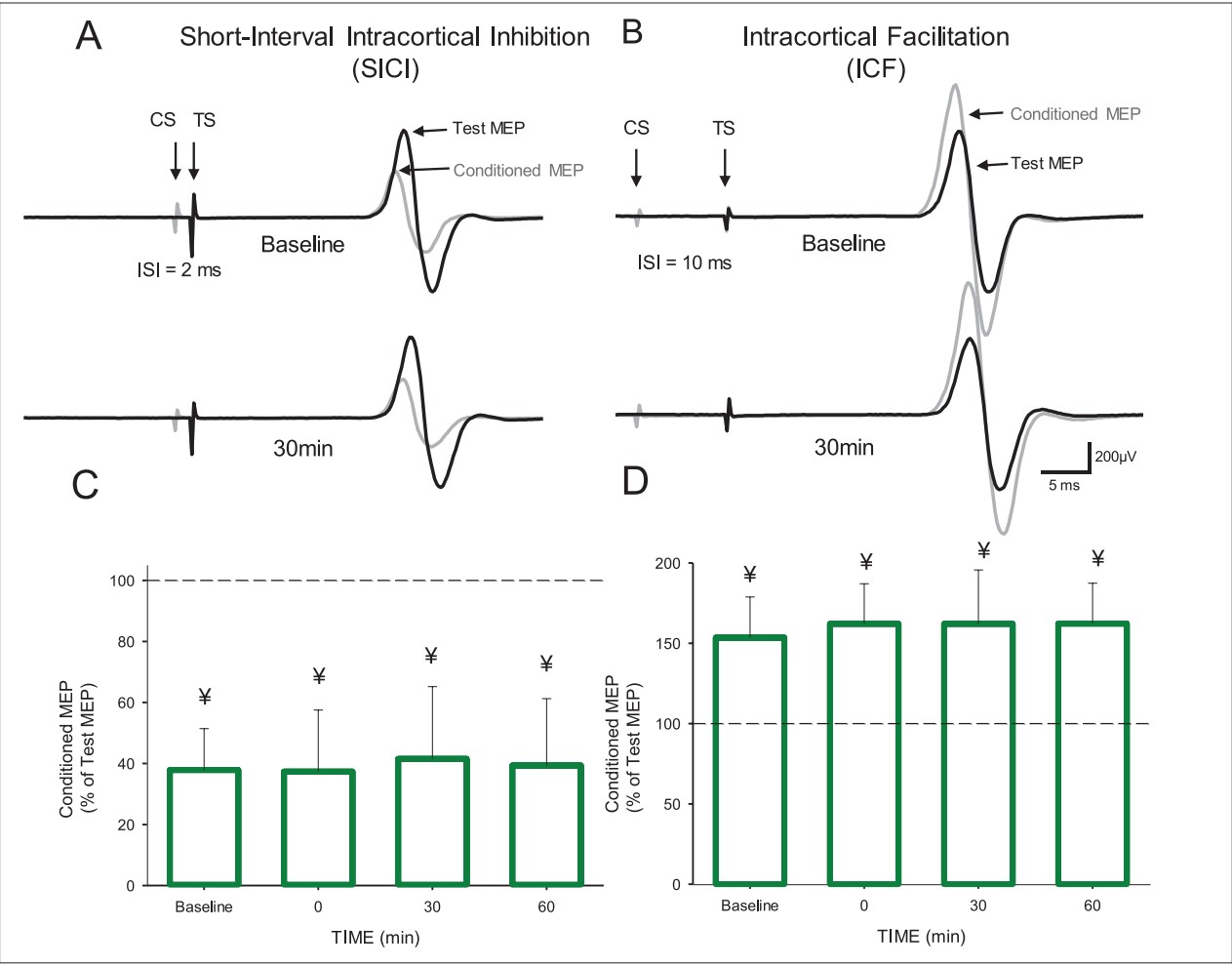

**Figure 3.** Short-interval intracortical inhibition (SICI) and intracortical facilitation (ICF). Raw traces showing SICI (**A**) and ICF (**B**) in the FDI muscle in a representative participant before and 30 min after the AIH protocol. Each waveform represents the average 15 test (black traces) and 15 conditioned (gray traces) MEPs. Arrows at the beginning of each trace indicate the test (TS) and conditioned (CS) stimuli used during testing. Arrows also indicate the test (black traces) and conditioned (gray traces) MEPs. Group SICI (n = 10, **C**) and ICF (n = 11, **D**) data before and after the AIH protocol. The abscissa shows the time of measurements (baseline, immediately after (0), 30, and 60 min after the AIH protocol). The ordinate shows the conditioned MEP as a percentage of the test MEP at baseline (% of Test Response). Error bars indicate SDs. ¥p<0.05, significant inhibition compared with the test response.

DOI: https://doi.org/10.7554/eLife.34304.003

Repeated-measures ANOVA showed no effect of TIME ($F_{3,9}$=0.6, p=0.6) on SICI. We found that the magnitude of SICI remained similar to baseline (37.9 ± 3.0%) immediately after (37.6 ± 4.5%), 30 (41.5 ± 5.3%) and 60 min (39.4 ± 6.9%) after the AIH protocol (*Figure 3C*). Similarly, repeated-measures ANOVA showed no effect of TIME ($F_{3,10}$=0.6, p=0.5) on ICF. The magnitude of ICF remained similar to baseline (53.5 ± 25.3%) immediately after (62.1 ± 24.9%), 30 (62.0 ± 33.5%) and 60 min (62.1 ± 25.2%) after the AIH protocol (*Figure 3D*).

## MEPs elicited by electrical stimulation

*Figure 4* illustrates traces of MEPs elicited by electrical stimulation in the FDI muscle from a representative subject before and after the AIH protocol. Note that the amplitude of MEPs evoked by electrical stimulation (subcortically evoked MEPs) increased after the AIH for up to 75 min. Repeated-measures ANOVA revealed an effect of TIME ($F_{6,12}$=7.3, p<000.1) on MEP amplitude. *Post hoc* tests showed that MEP amplitude increased after 15 (1.16 ± 0.9 mV; p=0.01), 30 (1.18 ± 0.9 mV; p=0.007), 45 (1.19 ± 0.8 mV; p<0.001), 60 (1. 43 ± 1.1 mV; p<0.001), and 75 (1.15 ± 0.9; p=0.006) min after AIH compared

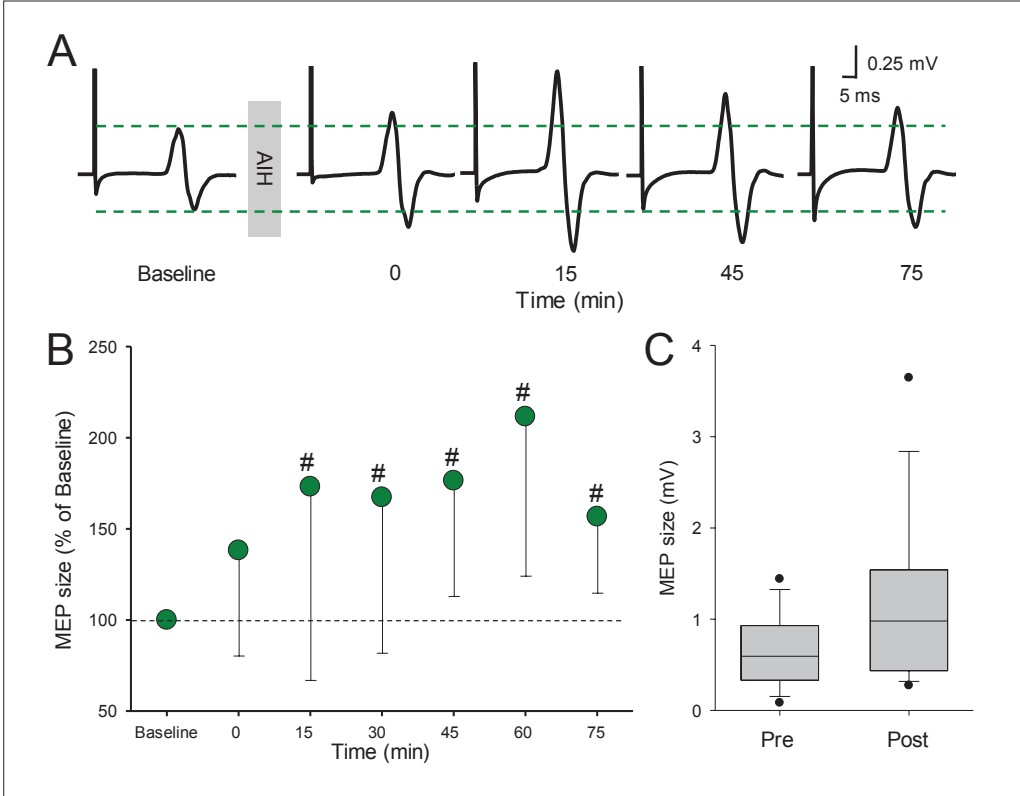

**Figure 4.** MEPs elicited by electrical stimulation. (**A**) Raw MEP traces elicited by electrical stimulation at the primary motor cortex in the FDI muscle in a representative participant before and after the AIH protocol. Each waveform represents the average of 10 MEPs. (**B**) Group data (n = 13). The abscissa shows the time of measurements (baseline, immediately after (0), 15, 30, 45, 60 and 75 min after the AIH protocol). The ordinate shows the MEP amplitude as a percentage of the MEP at baseline (% of Baseline). (**C**) Box-plots illustrate grouped MEP data before and after AIH (n=13). The abscissa shows the time of measurement (Pre and Post). Note that for post measurements data from 15 to 75 minutes were averaged.The ordinate shows the MEP amplitude (in millivolts). Whiskers illustrate 5th and 95th percentiles and dots represent minimal and maximal values. Error bars indicate SDs. #p<0.05, comparison with the baseline.

DOI: https://doi.org/10.7554/eLife.34304.004

with baseline (0.73 ± 0.4 mV; *Figure 4B*). As for MEPs evoked by TMS, we observed no changes in MEPs elicited by electrical stimulation immediately after the AIH protocol (p=0.37). Note that MEPs were facilitated in all subjects after the AIH protocol when comparing the average of tests conducted between 15 and 75 min after AIH with the baseline.

## F waves

Repeated-measures ANOVA revealed no effect of TIME ($F_{1,1}$=0.46, p=0.2), GROUP ($F_{1,18}$=0.466, p=0.5) or in their interaction ($F_{1,18}$=0.1, p=0.6) on M-max amplitude (*Figure 5*). Note that the M-max remained similar before and after AIH (baseline = 17.4 ± 3.6 mV, after = 17.8 ± 3.6; p=0.2) and sham AIH (baseline = 17.7 ± 3.5 mV, after = 17.9 ± 3.5 mV; p=0.3). Repeated-measures ANOVA showed no effect of TIME ($F_{6,11}$=0.3, p=0.8) on F-wave amplitude. The F-wave amplitudes were normalized to the amplitude of the M-wave recorded at the same stimulation intensity. The amplitude remained similar to baseline across time (baseline = 0.43 ± 0.4% of M-max; immediately after = 0.4 ± 0.3% of M-max; 15 min = 0.43 ± 0.4% of M-max; 30 min = 0. 46 ± 0.4% of M-max; 45 min = 0.44 ± 0.4% of M-max; 60 min = 0.41 ± 0.3% of M-max; 75 min = 0.43 ± 0.4% of M-max) after the AIH protocol. Repeated-measures ANOVA also showed no effect of TIME ($F_{6,11}$=0.5, p<0.7) on F-wave persistence. The F-wave persistence remained similar to baseline across time (baseline = 62.2 ± 25.8%; immediately after =

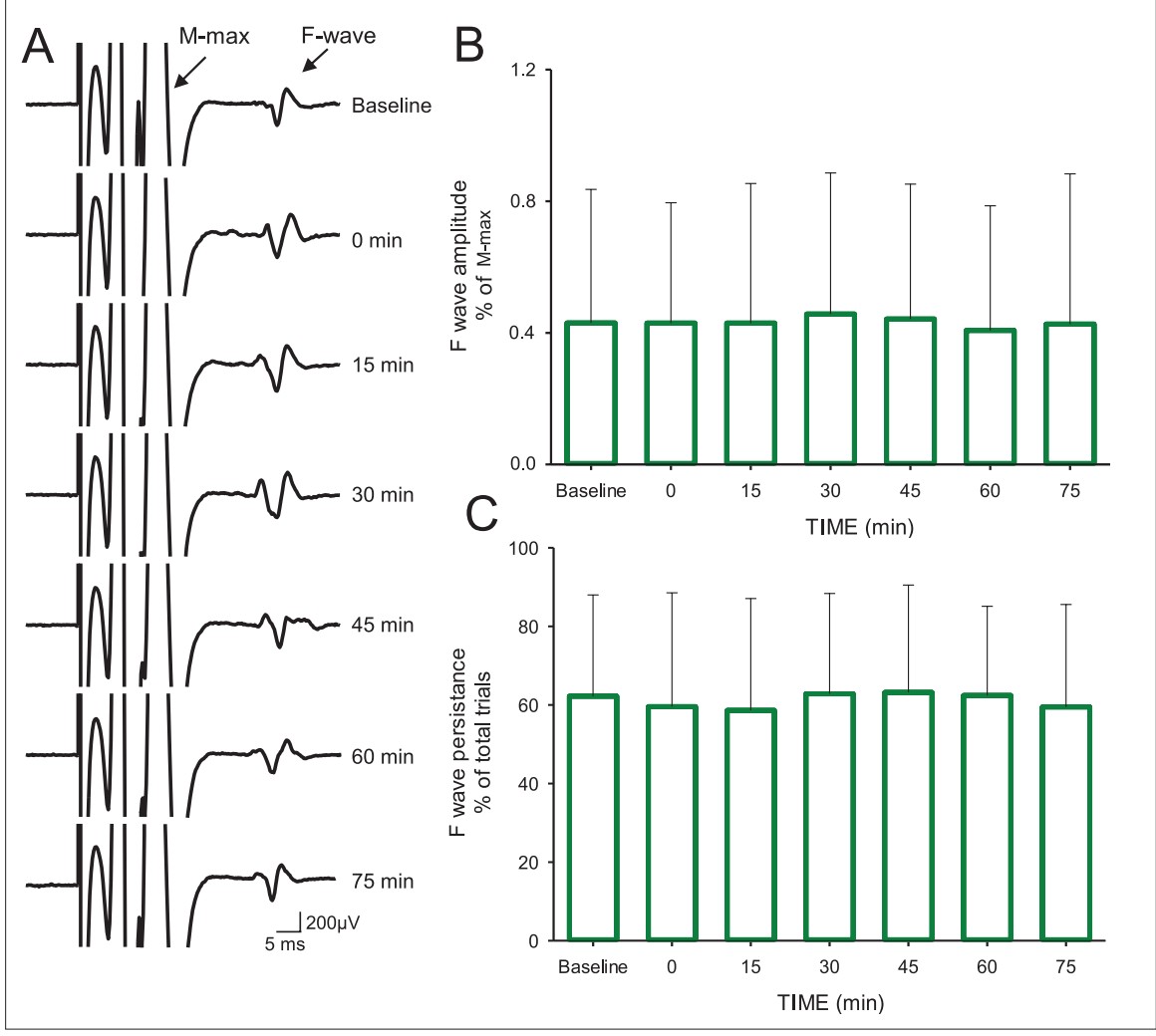

**Figure 5.** F-waves. (**A**) Maximal motor response (M-max) and F-wave traces recorded from the FDI muscle in a representative subject before and after the AIH protocol. The M-max is truncated to better visualize the F-wave. Note the multi-phasic shape of the M-max due to the post hoc filtering conducted to reduce the M-max 'tail' (see Materials and methods). Each waveform represents the average of 30 trials. Group data (n = 12) showing the F-wave amplitude (**B**) and persistence (**C**). In each graph, the abscissa shows the time of measurements (baseline, immediately after (0), 15, 30, 45, 60 and 75 min after the AIH protocol) and the ordinate shows the F-wave amplitude (expressed as a % of the M-max; **B**) and persistence (expressed as a % of total trials; **C**). Error bars indicate SDs.

DOI: https://doi.org/10.7554/eLife.34304.007

59.5 ± 28.9%, 15 min = 58.6 ± 28.5%, 30 min = 62.8 ± 25.5%, 45 min = 63.1 ± 27.3%, 60 min = 62.4 ± 22.7%; 75 min = 59.4 ± 26.1%) after the AIH protocol.

## Spike-timing dependent plasticity (STDP)

*Figure 6C* shows raw FDI MEPs traces from a representative participant. Note that MEP amplitude increased compared with baseline (gray traces) after the individual received AIH (green trace), STDP$_{AIH}$ (black trace) and STDP$_{sham\ AIH}$ (blue trace). Importantly, MEP amplitude increased to a larger extent when STDP was combined with AIH.

Repeated-measures ANOVA showed an effect of TIME ($F_{3,10}$=32.612, p<0.001), GROUP ($F_{2,10}$=8.06, p=0.003) and in their interaction ($F_{12,60}$=3.975, p=0.002) on FDI MEP amplitude. *Post hoc* tests showed that MEP amplitude increased to a larger extent after STDP$_{AIH}$ (77.2 ± 63.5%) compared with STDP$_{sham\ AIH}$ (36.72 ± 31.1%, p=0.005) and AIH (24.78 ± 31.1%, p=0.007) between 30 and 45 min after the protocol. Similarly, MEPs were larger between 60 and 75 min after the STDP$_{AIH}$ (78.6 ± 52.1%) compared with STDP$_{sham\ AIH}$ (23.8 ± 31.7%; p<0.001) and AIH protocol (38.1 ± 17.6%; p=0.009;

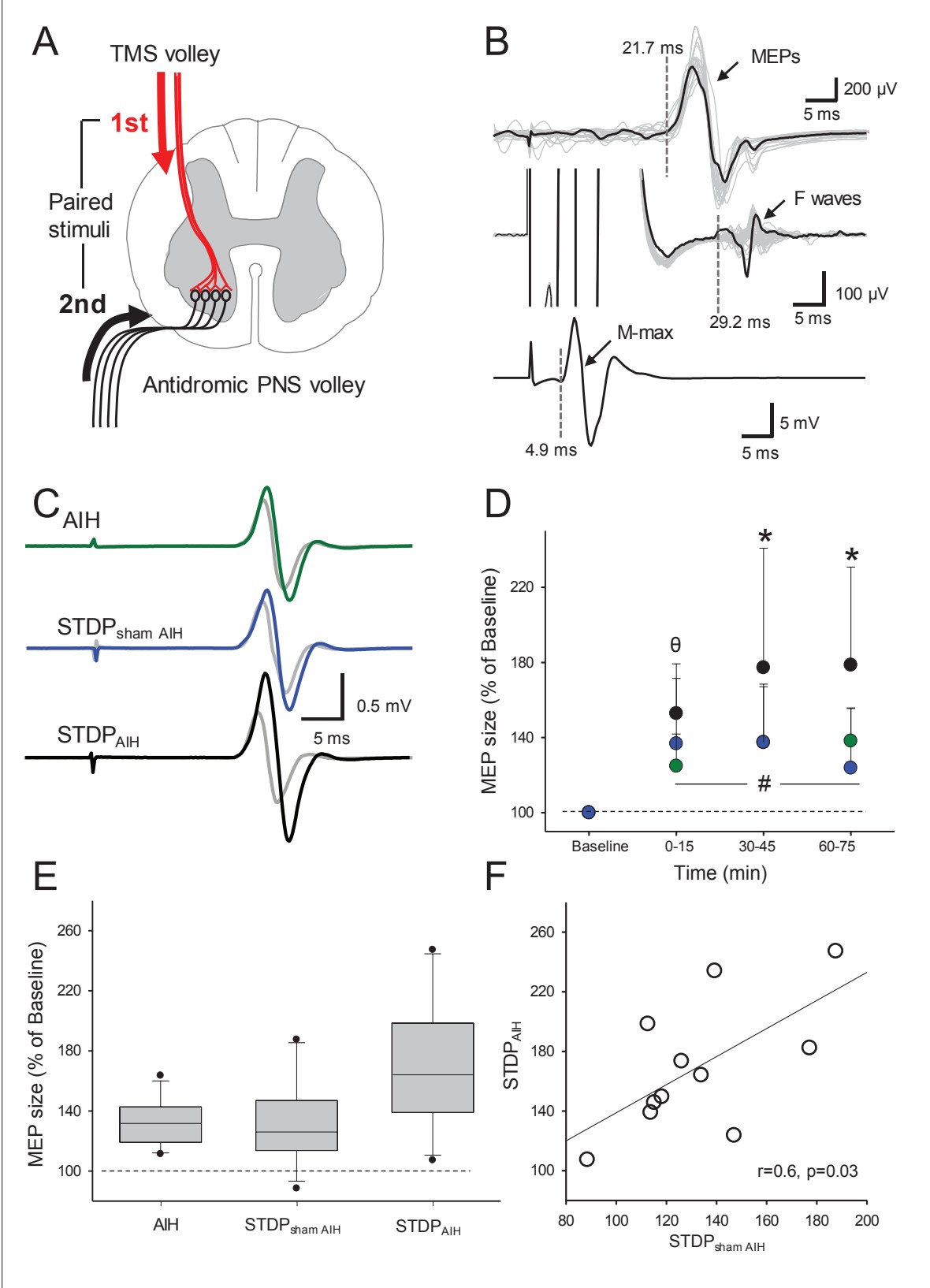

**Figure 6.** Spike-timing dependent plasticity (STDP). (**A**) Diagram showing the STDP protocol where corticospinal volleys evoked by TMS over the hand representation of the primary motor cortex were timed to arrive at corticospinal-motoneuronal synapses of the FDI muscle 2 ms before antidromic

*Figure 6 continued on next page*

*Figure 6 continued*

potentials elicited in motoneurons by peripheral nerve stimulation (PNS) of the ulnar nerve. (**B**) Raw traces showing a MEP, an F-wave, and the M-max from a representative subject recorded from the FDI muscle used to calculate central and peripheral conduction time to estimate the arrival of pre- and post-synaptic volleys at the corticospinal-motoneuronal synapse. (**C**) Raw MEP traces from a representative participant showing MEPs at baseline (grey traces) and immediately after AIH (green trace), STDP combined with sham AIH (STDP$_{sham\ AIH}$, blue trace) and STDP combined with AIH (STDP$_{AIH}$, black trace). Each trace is the average of 20 trials. (**D**) Group data (n = 11) showing the effect of AIH (green circles), STDP$_{sham\ AIH}$ (blue circles), and STDP$_{AIH}$ (black circles). The abscissa shows the time of measurements (baseline, immediately after (0–15), 30–45, and 60–75 min after each protocol) and the ordinate shows the MEP amplitude (expressed as a % of Baseline). (**E**) Graph shows box-plots group normalized MEP data at baseline (dotted line) and after the AIH and sham AIH protocols from all time points averaged. The abscissa shows the protocol tested (AIH, STDP$_{sham\ AIH}$ and STDP$_{AIH}$) and the ordinate shows the MEP amplitude (expressed as a % of Baseline). Whiskers illustrate 5th and 95th percentiles and dots represent minimal and maximal values. (**F**) Correlation analysis between increases in MEP amplitude after STDP$_{AIH}$ and STDP$_{sham\ AIH}$. Error bars indicate SDs. *p<0.05, STDP$_{AIH}$ different than AIH and STDP$_{sham}$,    p<0.05, STDP$_{AIH}$ different than AIH, #p<0.001, comparison with the baseline.

DOI: https://doi.org/10.7554/eLife.34304.005

*Figure 6D*). Note that MEP amplitude increased to a similar extent after STDP$_{sham\ AIH}$ and AIH immediately after (p=0.54), 30–45 min (p=0.99) and 60–75 min (p=0.17). Note that MEP amplitude increased after STDP$_{AIH}$ (77.9 ± 53.1%) to larger extent than the sum of the changes observed after STDP$_{sham\ AIH}$ and AIH (64.1 ± 32.2%) in most (8/11) participants. Also note, that the increases in MEP amplitude after STDP$_{AIH}$ correlated with increases in MEP amplitude after STDP$_{sham\ AIH}$ (r = 0.6, p=0.03; *Figure 6F*) but not after AIH (r = 0.2, p=0.4).

## Discussion

The novel finding in our study is that a single session of AIH enhances transmission in the human corticospinal pathway, likely related to corticospinal-motoneuronal synaptic plasticity. Specifically, we found that the amplitude of cortically evoked MEPs increased for 75 min after AIH but not sham AIH. Subcortically evoked MEPs increased while intracortical inhibition and facilitation remained unchanged after AIH, suggesting a subcortical origin for our effects. To further examine subcortical effects we tested STDP targeting spinal synapses and spinal motoneuron excitability and found that AIH increased STDP outcomes without changing motoneuronal excitability. We propose that AIH may present a novel approach for inducing spinal synaptic plasticity in the human corticospinal pathway.

### AIH effects on corticospinal excitability

Different factors increase the excitability of corticospinal neurons in humans including exercise, motor learning, repetitive stimulation of a peripheral nerve and/or cortical regions, and deafferentation (*Brasil-Neto et al., 1993*; *Tazoe and Perez, 2015*; *Chung et al., 2016*; *Berghuis et al., 2017*). Here, for the first time, we show that brief exposures to hypoxic air interspaced with periods of ambient room air for a total of 30 min (i.e. AIH) enhances corticospinal excitability for 75 min. This agrees with evidence showing that repeated AIH exposure in rats changes the neurochemical phenotype of corticospinal cells in ways that suggest enhanced function (*Satriotomo et al., 2016*). MEPs elicited by TMS, used to measure corticospinal excitability, likely reflect activation of corticospinal cells making direct and indirect (i.e. via interneurons), connections with spinal motoneurons (*Petersen et al., 2010*). This also agrees with effects of hypoxia seen in patients with chronic obstructive lung disease with arterial hypoxemia; these patients exhibit less inhibition in the primary motor cortex (*Oliviero et al., 2002*) . In contrast, exposure to sustained hypoxia for a similar amount of time as used in our study does not change corticospinal excitability (*Szubski et al., 2006*; *Goodall et al., 2010*; *Rupp et al., 2012*), highlighting that the duration (chronic versus acute) and pattern (intermittent versus sustained) are of importance.

An important question is what neuronal mechanisms contribute to increases in corticospinal excitability after a single AIH exposure? Since cortical and subcortical mechanisms could both contribute to changes in MEP amplitude (*Burke and Pierrot-Deseilligny, 2010*), it is possible that both sources influenced our results. To examine the contribution from cortical networks, we examined SICI by using a paired-pulse TMS paradigm (*Kujirai et al., 1993*). Studies using pharmacology and epidural recordings indicate that SICI likely reflects activation of γ-aminobutyric acid (GABA) inhibitory circuits (GABA$_A$) in the primary motor cortex (*Hanajima et al., 1998*; *Di Lazzaro et al., 2000*). Therefore, the

lack of changes in SICI after the AIH protocol suggests that it is less likely that changes in the primary motor cortex contributed to our results. Similarly, the lack of changes in ICF after AIH support a non-cortical site of plasticity. ICF is sensitive to changes in both GABAergic (*Ziemann et al., 1996a, b*) and glutamatergic transmission (*Liepert et al., 1997*) in the primary motor cortex. Collectively, our findings provide no evidence that increased MEP amplitude after AIH involves cortical mechanisms.

Interestingly, we found that the amplitude of MEPs elicited by electrical stimulation of corticospinal axons increased after AIH. Electrical stimulation of the primary motor cortex likely activates axons of pyramidal tract neurons in the subcortical white matter at the axon initial segment (*Day et al., 1989*) whereas electrical stimulation of the cervicomedullary junction activates corticospinal axons directly and faraway from the cortex (*Taylor and Gandevia, 2004*). Thus, a possible interpretation of our results is that changes in MEPs evoked by electrical stimulation after AIH have a subcortical origin. This is supported by the parallel time course of changes in the amplitude of MEPs elicited by both TMS and electrical stimulation after AIH. Further supporting this idea, rat studies demonstrate that even a single AIH exposure changes the connectivity of spinal interneurons that may project to relevant motoneurons (*Streeter et al., 2017*). To further examine the origin of subcortical effects, we measured transmission at spinal synapses and the excitability of spinal motoneurons. In human upper-limb muscles, spinal synapses have been targeted noninvasively by using paired corticospinal-moto-neuronal stimulation, where corticospinal volleys evoked by TMS over the primary motor cortex are timed to arrive at corticospinal-motoneuronal synapses before or after antidromic potentials evoked in motoneurons by electrical stimulation of a peripheral nerve; this protocol elicits STDP-like changes (*Taylor and Martin, 2009*; *Bunday and Perez, 2012*; *Fitzpatrick et al., 2016*; *Urbin et al., 2017*).

Here, we found that combined STDP and AIH caused larger increases in corticospinal excitability versus AIH alone or STDP combined with sham AIH. The facilitatory effect of AIH and STDP was larger than the sum of the changes caused by each protocol alone in the majority if participants, suggesting that these forms of plasticity share some overlapping mechanisms. STDP is thought to engage long-term potentiation (LTP)-like mechanisms that depend on N-methyl-D-aspartate (NMDA) (*Bi and Poo, 1998*). AIH-induced phrenic motor facilitation requires NMDA receptor activation to maintain (versus initiate) plasticity (*Fuller et al., 2000*; *McGuire et al., 2005*, *2008*). In humans, STDP-like changes elicited through repeated noninvasive stimulation is blocked by dextromethorphan, a drug that antagonizes NMDA receptors (*Dongés et al., 2018*). Note that in our study, we followed up the effect of AIH on corticospinal excitability in one participant and found that corticospinal responses returned to baseline ~120 hr after a single AIH session, consistent with previous results on ankle strength in people with spinal cord injury (*Trumbower et al., 2012*) and AIH-induced long-term facilitation of phrenic nerve activity in rats (*Bach and Mitchell, 1996*; *Nichols et al., 2012*). Measurements of intracortical excitability after AIH, such as SICI and ICF, also reflect NMDA receptor dependent mechanisms operating at the cortical level (*Ziemann et al., 1998*; *Schwenkreis et al., 1999*). Thus, the lack of change in these outcomes highlights the specificity of STPD and AIH at the spinal level.

An unexpected result was the lack of changes in the amplitude and persistence of F-waves after AIH because it is an indirect measure of motoneuronal excitability. Although some limitations have been described in the extent to which F-waves measure motoneuron excitability (*Hultborn and Nielsen, 1995*; *Espiritu et al., 2003*), this outcome measure is adequately sensitive to detect changes in motoneuron excitability in some experimental paradigms (*Giesebrecht et al., 2011*; *Khan et al., 2012*; *Rossi et al., 2012*). Episodic release of serotonin from raphe spinal projections to the cervical spinal cord is necessary and sufficient for induction but not maintenance of AIH-induced long-term facilitation in phrenic motoneurons (*Fuller et al., 2001*). In humans, increased extra-cellular serotonin levels caused by the selective serotonin reuptake inhibitor citalopram enhance and prolong NMDA-mediated corticospinal plasticity (*Nitsche et al., 2009*); however, the role of serotonin on STDP or AIH effects is unknown and it will be interesting to examine in future studies. On one hand, evidence that AIH acts subcortically, but does not change motoneuron excitability is consistent with the conclusion that AIH acts on synapses of the corticospinal pathway onto moto-neurons per se. This conclusion is consistent with known effects of AIH on phrenic motor plasticity, which appear to arise within phrenic motoneurons, strengthening motoneuron synaptic inputs from medullary respiratory premotor neurons (*Dale-Nagle et al., 2010*; *Devinney et al., 2013*; *Dale et al., 2017*). On the other side, it is important to consider that even though motor units of all sizes could contribute to F-wave activity (*Dengler et al., 1992*), F-waves are likely to be generated

by large motoneurons because small motoneurons can generate H-reflexes, which prevents these motor units to contribute to F-waves (*Espiritu et al., 2003*). TMS recruits motor units in an orderly fashion involving the recruitment of small and large motoneurons (*Bawa and Lemon, 1993*). Therefore, changes in the excitability of motoneurons, although not detected here, cannot be completely excluded.

### Functional significance

A growing number of studies supports the potential of AIH to enhance motor recovery in humans with disabilities due to spinal cord injury (*Trumbower et al., 2012*; *Hayes et al., 2014*; *Lynch et al., 2017*; *Navarrete-Opazo et al., 2017a*). Here, we provide evidence for a physiological mechanism that can contribute to those effects in humans. Plasticity in the corticospinal pathway contributes to improvements in motor performance in humans with and without spinal cord injury (*Oudega and Perez, 2012*). Importantly, STDP-like changes has been effective in modulating corticospinal excitability and voluntary motor output in humans with chronic spinal injuries (*Bunday and Perez, 2012*; *Urbin et al., 2017*). Parallel increases in corticospinal and subcortical excitability, and STDP-like changes, suggest that corticomotoneuronal spinal synapses are primary loci for the underlying neuroplasticity. This is important because interventions that successfully engage synaptic plasticity following CNS damage remain limited.

## Materials and methods

### Subjects

Twenty-one able-bodied control participants took part in the study (mean age 27.7 ± 6.9 years; eight female, four left-handed) participated in the study. The experimental procedures were approved by the Institutional Review Board at the University of Miami and were in accordance with the declaration of Helsinki. All participants signed a written informed consent before the first testing session.

### Electromyographic (EMG) and force recordings

Electromyography was recorded from the first dorsal interosseous (FDI) of the dominant side through surface electrodes (Ag–AgCl, 10 mm diameter) secured to the skin in a muscle tendon montage with one electrode placed over the belly of the muscle. All signals were amplified (x200), filtered (30–1000 Hz), and sampled at 4 kHz for off-line analysis (Cambridge Electronic Design 1401 with Signal software v4.1). At the start of the experiment, subjects were instructed to perform 2–3 brief maximal voluntary contraction (MVCs) for 3–5 s into index finger abduction, separated by ~30 s of rest. During MVCs, force exerted at the proximal interphalangeal joint of the index finger was measured by load cells (Honeywell, Ltd., range ±498.1 N, voltage ±5 V, high-sensitivity transducer 0.04 V/N). Force was sampled at 200 Hz and stored on a computer for off-line analysis.

### Experimental paradigm

Nineteen subjects participated in two randomized blinded testing sessions separated by at least one week while seated with both arms flexed at the elbow by 90°. In one session, AIH was delivered through a mask with two one-way valves restricting inspiration to the top valve and expiration to the bottom valve using an oxygen generator (Hypoxico Inc, EVEREST SUMMIT II, New York; *Figure 1A*). The hypoxic ($F_iO$ = 0.09 ± 0.001) and normoxic ($F_iO_2$ = 0.2) gas mixture was delivered through the top valve. The AIH protocol consisted of 15 episodes of hypoxia/normoxia interspaced with 1 min of inspiring ambient air (*Figure 1B*). In the other session, sham AIH was delivered as described above using only normoxic gas. We monitored oxygen concentration before, during, and after both sessions using a Handi+ oxygen monitor (maxtec, Salt Lake City, Utah, US). $S_pO_2$ levels were different throughout both sessions (AIH = 81.3 ± 1.2, sham AIH = 97.9 ± 0.3, p<0.001; *Figure 1D*). Participants breathed through the mask for 2 min before the AIH and sham AIH for familiarization with the procedures. $S_pO_2$ and heart rate were taken using a pulse oximeter that stores data with a 4 s data collection rate (Nonin 3150 WristOx2 , Nonin medical Inc, Plymouth, Minnesota, United States).

## TMS

Transcranial magnetic stimuli were applied using a figure-of-eight coil (loop diameter 70 mm) through a Magstim 2002 magnetic stimulator (Magstim, Whitland, Dyfed, UK) with a monophasic current waveform. We determined the optimal position for eliciting a MEP in the FDI muscle (hot spot) by moving the coil, with the handle pointing backward and 45° away from the midline, in small steps along the hand representation of the primary motor cortex. The hot spot was defined as the region where the largest MEP in the FDI could be evoked with the minimum intensity (*Rothwell et al., 1999*). With this coil position, the current flowed in a posterior-anterior direction and probably produced D and early I wave activation (*Sakai et al., 1997*). The TMS coil was held to the head of the subject with a custom coil holder, while the head was firmly secured to a headrest by straps. Coil position was monitored using a frameless stereotaxic neuro-navigation system (Brainsight 2, Rogue Research, Montreal, Canada). MEPs elicited by TMS and electrical stimulation were measured before, immediately after (0), and 15, 30, 45, 60, and 75 min after the AIH and sham AIH protocol (*Figure 1C*). Note that in one participant we followed up the effects of AIH until MEP amplitude returned to baseline. To examine the mechanisms contributing to changes in MEP amplitude observed after the AIH protocol we tested MEPs evoked by electrical stimulation and F-waves at similar time points. Due to the adjustments needed, SICI and ICF were tested immediately after (0), and 30, and 60 min after the AIH protocol. We stopped testing at 60 min since no changes were observed at any of the time points examined.

## MEPs elicited by TMS

The resting motor threshold (RMT) was defined as the minimal stimulus intensity required to induce MEPs >50 µV peak-to-peak amplitude above the background EMG in 5/10 consecutive trials in the relaxed muscle (*Rothwell et al., 1999*). MEPs were tested at rest (AIH = 1.10 ± 0.79 mV, sham AIH = 1.15 ± 0.61 mV; p=0.83) at intensities of 120% of RMT in both sessions (AIH = 55.7 ± 13.6% of maximal stimulator output, MSO; sham AIH = 54.2 ± 13.4% of MSO; p=0.1). The MEP-max was defined at rest by increasing stimulus intensities in 5% steps of maximal device output until the MEP amplitude did not show additional increases (4.5 ± 2.3 mV). At baseline, 60 TMS pulses were delivered at 4 s intervals in sets of 20 separated by resting periods as needed. Two sets of 20 MEPs were collected at each time point following AIH and sham AIH. MEPs were visually inspected and analyzed trial by trial.

## SICI

SICI was tested to examine excitability of intracortical inhibitory circuits. SICI was tested at rest using a previously described method (*Kujirai et al., 1993*); n = 10. A conditioning stimulus (CS) was set at an intensity needed to elicit ~50% of SICI, which corresponded to ~80% of RMT (36.5 ± 5.9). The same stimulation intensity was used before and after the AIH protocol. Because MEP amplitude increased after the AIH protocol the test stimuli (TS) was set at an intensity needed to elicit an MEP ~1 mV before (1.04 ± 0.2 mV), immediately after (1.0 ± 0.3 mV), 30 (1.08 ± 0.3 mV), and 60 min (1.06 ± 0.2 mV) after the AIH protocol. This was accomplished by measuring SICI and decreasing the intensity of the TS as needed in order to elicit a test MEP as close as possible to the baseline MEP. SICI was calculated by expressing the amplitude of conditioned MEP as a % of the test MEP amplitude [(conditioned MEP x 100)/(test MEP)]. Two sets of 20 test MEPs and 20 conditioned MEPs were tested at each time point.

## ICF

We examined activity of intracortical excitatory circuits by measuring ICF. We used a previously described method to test ICF (*Kujirai et al., 1993*); n = 11. A CS was set at an intensity needed to elicit ~50% of ICF, which corresponded to ~89% of RMT (43.1 ± 11.9). The same stimulation intensity was used before and after the AIH protocol. The TS was set at an intensity needed to elicit an MEP ~1 mV before (1.1 ± 0.2 mV), immediately after (1.1 ± 0.2 mV), 30 (1.1 ± 0.3 mV), and 60 min (1.1 ± 0.2 mV) after the AIH protocol. The CS was delivered 10 ms before the TS. We also tested ICF by adjusting the amplitude of the test MEP to match MEP amplitudes produced at baseline. ICF was calculated by expressing the amplitude of conditioned MEP as a % of the test MEP amplitude [(conditioned MEP x 100)/(test MEP)]. Two sets of 15 test MEPs and 15 conditioned MEPs were acquired at each time point.

## MEPs elicited by electrical stimulation

We measured corticospinal transmission and/or motoneuron excitability by stimulating the cortico-spinal tract using high voltage electrical current (200 µs duration, Digitimer DS7AH) passed between two small gold-cup electrodes (GRASS Technologies , Astro Med , Inc., Warwick, Rhode Island, USA) placed behind the mastoid process at the cervicomedullary level (CMEPs, n = 6; *Taylor and Gandevia, 2004*). The stimulation intensity was adjusted to produce a response of ~3% M-max (258.3 ± 39.8 mA). Cervical root activation was investigated by increasing the intensity until an abrupt decrement in latency occurred, then turning the intensity down and by verifying that the response was potentiated by a small background contraction (*Taylor, 2006*). The latency of CMEPs (19.5 ± 1.3 ms) was shorter than the latency of MEPs elicited by TMS (22.6 ± 1.4 ms, p=0.002). In seven subjects, the cortico-spinal tract was stimulated through the scalp with the cathode at the vertex and anode 7 cm lateral towards the external meatus acusticus (*Day et al., 1989*) at an intensity eliciting responses of ~3% of M-max in the FDI muscle (211.9 ± 48.4 mA). The latency of MEPs elicited by cortical electrical stimu-lation (21.3 ± 2.0 ms) was shorter than the latency of MEPs evoked by TMS (23.6 ± 2.0 ms, p=0.007), suggesting that we stimulated corticospinal axons directly. Because MEPs elicited by cervicomedul-lary and cortical electrical stimulation showed the same changes across conditions we grouped all results together under MEPs elicited by electrical stimulation. Five to ten MEPs elicited by electrical stimulation were tested at each time point.

## F-waves

To make inferences about changes in motoneuron excitability we tested F-waves amplitude and persistence. F-waves were measured using supramaximum stimulus intensity to the ulnar nerve at the wrist (200 µs pulse duration, DS7A; Digitimer) with a monopolar bar electrode with the cathode positioned proximally (n = 12). The stimuli were delivered at 1 Hz at an intensity of 150% of the maximal motor response (M-max). For each trial, we quantified each F-wave's peak-to-peak ampli-tude (expressed relative to the M-max) and F-wave persistence (the percentage of stimuli evoking a response; *McNeil et al., 2013*). F-wave trials were filtered using a 2nd order Bessel high-pass filter (200 Hz) to 'flatten the tail of the M-wave' (*Khan et al., 2012*). M-max was acquired using our regular filter parameters (30–1000 Hz). An F-wave was considered to be present if a response with a proper latency (minimum of 20 ms) had an amplitude $\geq$20 µV above background. If an F-wave was not present, an amplitude of '0' was included in the mean (*Butler and Thomas, 2003*). The same result was observed whether '0' F-wave amplitudes were included in the analysis or not. Sixty F-waves were tested at each time point.

## STDP

We tested the STDP protocol (*Taylor and Martin, 2009*; *Bunday and Perez, 2012*) with either AIH (STDP$_{AIH}$) or sham AIH (STDP$_{sham\ AIH}$) in two additional randomized sessions separated by at least one week (n = 11). During STDP, 180 pairs of stimuli were delivered every 10 s (~34 min, 0.1 Hz) where corticospinal volleys evoked by TMS over the hand representation of the primary motor cortex were timed to arrive at corticospinal-motoneuronal synapses of the FDI muscle 2 ms before antidromic potentials evoked in motoneurons by peripheral nerve stimulation (PNS) of the ulnar nerve. Magnetic stimuli were delivered with 100% of stimulator output. PNS was set at an intensity of 150% of the intensity needed to evoke M-max. The cathode was place most proximal to avoid a potential anodal blockade of the antidromic volley and thereby ensure that these were indeed supramaximal. Inter-stimulus intervals (ISIs) between TMS and PNS were tailored to individual subjects based on central (CCT) and peripheral (PCT) conduction times calculated from latencies of MEPs, F-wave, and M-max. MEP latencies were recorded while the index finger performed 10% of isometric MVC in the abduction direction. The onset latency was defined as the time when each response exceeded 2 SD of the mean rectified pre-stimulus activity (100 ms) in the averaged wave-form. For F-waves, the earliest onset latency was used for calculations. PCT was calculated using the following equation: (F-wave latency – M-max latency) * 0.5 and CCT was calculated using the following equation: MEP latency – (PCT +M-max latency). Two sets of 20 MEPs were collected at each time point.

## Statistics

Normal distribution was examined using the Shapiro-Wilk's test and Mauchly's test was used to test sphericity. Using SPSS, data were log transformed when they were not normally distributed as found on MEPs elicited by TMS, $STDP_{AIH}$ and F-waves. Greenhouse-Geisser correction statistics were used to reveal significant F values when sphericity could not be assumed (*Armstrong, 2017*). Repeated-measures analysis of variance (ANOVA) was performed to examine the effect of TIME (Baseline, 0, 15, 30, 45, 60 and 75 min) and GROUP (AIH and sham AIH) effects on MEP amplitude normalized to baseline values. The same analysis was used to determine the effect of TIME (different time points tested) on SICI, ICF, MEPs elicited by electrical stimulation, and F-wave amplitude and persistence. We also measured the effect of STDP ($STDP_{AIH}$ and $STDP_{sham\ AIH}$) on MEP amplitude and the effect of TIME and GROUP on $S_pO_2$ levels. Paired t-tests were used to compare MEP amplitude and intensity at baseline during AIH and sham protocols, and latencies of MEPs evoked by electrical stimulation and TMS. Tukey's honestly significant difference *post hoc* test was used to test for significant comparisons. Significance was set at $p < 0.05$. Group data are presented as the mean ±SD in the text.

## Additional information

### Funding

| Funder | Grant reference number | Author |
| --- | --- | --- |
| National Institutes of Health | R01NS090622 | Monica A Perez |
| U.S. Department of Veterans Affairs | Veteran Affairs Merit Review I01RX000815 | Monica A Perez |
| U.S. Department of Veterans Affairs | Veteran Affairs Merit Review I01RX002474 | Monica A Perez |

The funders had no role in study design, data collection and interpretation, or the decision to submit the work for publication.

### Author contributions

Lasse Christiansen, Conceptualization, Resources, Data curation, Software, Formal analysis, Supervision, Funding acquisition, Validation, Visualization, Methodology, Writing – original draft, Project administration, Writing – review and editing; MA Urbin, Conceptualization, Methodology, Project administration, Investigation; Gordon S Mitchell, Conceptualization, Resources, Software, Supervision, Validation, Visualization, Methodology, Writing – original draft, Writing – review and editing, Investigation; Monica A Perez, Conceptualization, Resources, Data curation, Software, Formal analysis, Supervision, Funding acquisition, Validation, Visualization, Methodology, Writing – original draft, Project administration, Writing – review and editing, Investigation

### Author ORCIDs

Lasse Christiansen http://orcid.org/0000-0002-9316-1529
Monica A Perez http://orcid.org/0000-0002-8676-3492

### Ethics

The experimental procedures were approved by the local ethics committee at the University of Miami, were in accordance with the declaration of Helsinki and all participants gave their written informed consent before the first test session.

### Decision letter and Author response

Decision letter https://doi.org/10.7554/eLife.34304.009
Author response https://doi.org/10.7554/eLife.34304.010

# Additional files

## Supplementary files

- Transparent reporting form.

DOI: https://doi.org/10.7554/eLife.34304.008

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
