## [Decision Letter]

Thank you for submitting your article "Acute Intermittent Hypoxia Enhances Corticospinal Synaptic Plasticity in Humans" for consideration by*eLife*. Your article has been favorably evaluated by Sean Morrison (Senior Editor) and three reviewers, one of whom, Jan-Marino Ramirez (Reviewer #1), is a member of our Board of Reviewing Editors. The following individual involved in review of your submission has agreed to reveal their identity: Christopher Wilson (Reviewer #2).

The reviewers have discussed the reviews with one another and the Reviewing Editor has drafted this decision to help you prepare a revised submission.

Summary:

The study by Christiansen et al. is conceptually an important advance as it expands our understanding of the effects of intermittent hypoxia on subcortical pathways in humans.

It has long been known that intermittent hypoxia results in long-term plasticity that involves changes in a variety of systems including the respiratory motor output and the central respiratory network located within the preBötzinger complex. Much has been learned about the underlying intracellular pathways. Moreover, these findings have recently been adapted to improve spinal circuitries after spinal cord injury in humans.

The important question is whether acute intermittent hypoxia (AIH) can also alter descending corticospinal pathways and whether these alterations contribute to the observed changes at the spinal level. Cortical plasticity plays a critical role in the recovery from spinal cord injury in humans, but this has not been studied in the context of AIH. The authors find significant changes, yet, the observed effects do not originate in the cortex, but rather in subcortical structures. Moreover, and probably not surprisingly, the plasticity discovered by the authors is different from the long-term plasticity seen at the spinal level and lasts only for 75 minutes. This is an interesting and conceptually important finding, as it suggests that these descending pathways likely involve different mechanistic pathways and a different sensitivity to AIH. Indeed, this may be adaptive, since it may be detrimental if such pathways are affected in a long-term manner by brief exposure to intermittent hypoxia -note the exposure lasted only for 30 minutes.

Essential revisions:

1) The authors need to explicitly point out the differences between the modulation of the descending corticospinal pathways when compared to the already known metaplasticity in spinal cord networks, which typically last for a relatively long time. There is not a single mention of the fall-off of MEP amplitude at 75 minutes in Figure 2 (MEPs elicited by TMS) and in Figure 4 (MEPS elicited by electrical stim). If the response is truly a potentiated form of drive to the peripheral musculature, then it should last more than 75 minutes to have any usefulness in a clinical setting.

Did the authors try longer lasting stimulations (<30 mins)? Are they afraid of detrimental consequences? Moreover, the argument that spike-timing dependent is the key mechanism by which AIH alters the peripheral motor circuitry seems to be hobbled by this fall off in effect at 75 minutes. While I think the authors' assertion that the potentiation is peripheral is reasonable based on their data, discussion of the time course of effect and making sure that the 75 min data for TMS and electrical stim is*not*averaged into their results will be important.

2) The statistical analyses are described opaquely. Which data were not normally distributed and then log-transformed? Provide a reference for your use of Greenhouse-Geisser correction because it is not clear which of your data violates sphericity or how that was assessed. Please report which data violated these assumptions, perhaps with a comment about why these problems in the data are present. What statistical analysis software was used? Why was Bonferroni's used (which is unnecessarily conservative) as the post hoc correction method instead of something else (Tukey's HSD for example)?

3) The manuscript would be more easily accessible to readers outside the specific area of research if motivations for the SISI, ICF, MEPs elicited by electrical stimulation, F wave, and STDP tests are provided in Results. I recommend that the authors add a sentence or two to the beginning of each of these sections that describe the purpose of these tests. Although the rationales are addressed in the Discussion, earlier descriptions will be helpful to readers not familiar with these techniques.

---

## [Author Response]

Essential revisions:1) The authors need to explicitly point out the differences between the modulation of the descending corticospinal pathways when compared to the already known metaplasticity in spinal cord networks, which typically last for a relatively long time. There is not a single mention of the fall-off of MEP amplitude at 75 minutes in Figure 2 (MEPs elicited by TMS) and in Figure 4 (MEPS elicited by electrical stim). If the response is truly a potentiated form of drive to the peripheral musculature, then it should last more than 75 minutes to have any usefulness in a clinical setting.Did the authors try longer lasting stimulations (<30 mins)? Are they afraid of detrimental consequences? Moreover, the argument that spike-timing dependent is the key mechanism by which AIH alters the peripheral motor circuitry seems to be hobbled by this fall off in effect at 75 minutes. While I think the authors' assertion that the potentiation is peripheral is reasonable based on their data, discussion of the time course of effect and making sure that the 75 min data for TMS and electrical stim is not averaged into their results will be important.

In this thoughtful comment, we will clarify two different issues. First, the MEP amplitude did not fall-off at 75 min post-AIH. We simply stopped measuring electrophysiological outcomes at that point in all subjects. The second related clarification is that this is an investigation of AIH-induced plasticity, which is more akin to the phrenic nerve response following a single presentation of AIH known as phrenic long-term facilitation. Both corticospinal plasticity following a single AIH presentation and phrenic LTF follow similar time courses post-AIH (lasting at least 75 minutes). We followed up in one participant, and found that the corticospinal facilitation presented here returned to baseline ~120 minutes post-AIH. This time course is slightly shorter, but is consistent with the time course of AIH effects on leg strength in people with chronic spinal cord injury (Trumbower et al., 2012; Lynch et al., 2016), and is similar to the time course of phrenic LTF (many studies).

When AIH is presented repeatedly for days (i.e. repetitive AIH preconditioning), it is true that phrenic long-term facilitation exhibits metaplasticity (i.e. enhanced plasticity in response to a single AIH presentation; Fields and Mitchell, 2015; MacFarlane et al., 2017). When humans with spinal injury are exposed to repeated AIH for 5 days, the impact is similarly enhanced, particularly when combined with walking practice (Hayes et al., 2014; Navarrete-Opazo et al., 2016; Navarrete-Opazo et al., 2017). This was not the topic of study described in the present manuscript. However, we would like to emphasize that the translation of AIH to therapeutic applications was the direct result of initial, mechanistic (basic science) studies of phrenic long-term facilitation. Similar studies of mechanisms giving rise to limb/digit facilitation induced by AIH have simply not been done. Thus, the present study represents an important contribution to the literature: this is essentially the first verification that AIH potentiates corticospinal responses in a limb/finger, and that the site of plasticity involves potentiation somewhere between the brainstem and the synapse onto the relevant motoneurons. Further, we demonstrate that a form of activity-dependent plasticity (STDP) is at least additive with AIH effects, a novel finding in any model of motor plasticity. The following information was added to the manuscript:

“In humans, STDP-like plasticity elicited through repeated noninvasive stimulation is blocked by dextromethorphan, a drug that antagonizes NMDA receptors (Dongés et al., 2016) and its effects last for a few hours (Bunday and Perez, 2012). Note that in our study, we followed up the effect of AIH on corticospinal excitability in one participant and found that corticospinal responses returned to baseline ~120 hours after a single AIH session, consistent with previous results on ankle strength in people with spinal cord injury (Trumbower et al., 2012) and AIH-induced long-term facilitation of phrenic nerve activity in rats (Bach and Mitchell, 1996; Nichols et al., 2012).”

2) The statistical analyses are described opaquely. Which data were not normally distributed and then log-transformed? Provide a reference for your use of Greenhouse-Geisser correction because it is not clear which of your data violates sphericity or how that was assessed. Please report which data violated these assumptions, perhaps with a comment about why these problems in the data are present. What statistical analysis software was used? Why was Bonferroni's used (which is unnecessarily conservative) as the post hoc correction method instead of something else (Tukey's HSD for example)?

We clarified that data on MEPs elicited by TMS, STDP_AIH_and F-waves were not normally distributed and were log transformed. The log transformed MEP data elicited by TMS violated the sphericity assumption. Therefore, we conducted the analysis using SPSS (Split-plot/mixed-design), and applied the Green-Geisser correction after verifying the violation with Mauchly’s test

(Armstrong, 2017). We agree with the reviewers and now we use Tukey post hoc test. The following information was added to the manuscript:

“Using SPSS, data were log transformed when they were not normally distributed; thus, statistical analyses of MEPs elicited by TMS, STDP/AIH and F-waves were analysed on log transformed data. Greenhouse-Geisser correction statistics was used on MEPs elicited by TMS when sphericity could not be assumed (Armstrong, 2017).”

“Tukey's honestly significant difference (HSD) post hoc test was used to test for significant comparisons.”

3) The manuscript would be more easily accessible to readers outside the specific area of research if motivations for the SISI, ICF, MEPs elicited by electrical stimulation, F wave, and STDP tests are provided in Results. I recommend that the authors add a sentence or two to the beginning of each of these sections that describe the purpose of these tests. Although the rationales are addressed in the Discussion, earlier descriptions will be helpful to readers not familiar with these techniques.

These descriptions were added to the manuscript.